# Neem Leaf Extract Exhibits Anti-Aging and Antioxidant Effects from Yeast to Human Cells

**DOI:** 10.3390/nu16101506

**Published:** 2024-05-16

**Authors:** Jinye Dang, Gongrui Zhang, Jingjing Li, Libo He, Yi Ding, Jiaxiu Cai, Guohua Cheng, Yuhui Yang, Zhiyi Liu, Jiahui Fan, Linfang Du, Ke Liu

**Affiliations:** 1Key Laboratory of Bio-Resource and Eco-Environment of Ministry of Education, College of Life Sciences, Sichuan University, Chengdu 610065, China; 2College of Food and Biological Engineering, Chengdu University, Chengdu 610106, China

**Keywords:** *Azadirachta indica*, aging, antioxidant, catalase, *Saccharomyces cerevisiae*, chronological lifespan, network pharmacology

## Abstract

Neem leaves have long been used in traditional medicine for promoting longevity. However, the precise mechanisms underlying their anti-aging effects remain elusive. In this study, we investigated the impact of neem leaf extract (NLE) extracted from a 50% ethanol solution on the chronological lifespan of Saccharomyces cerevisiae, revealing an extension in lifespan, heightened oxidative stress resistance, and a reduction in reactive oxygen species. To discern the active compounds in NLE, LC/MS and the GNPS platform were employed. The majority of identified active compounds were found to be flavonoids. Subsequently, compound-target pharmacological networks were constructed using the STP and STITCH platforms for both S. cerevisiae and Homo sapiens. GOMF and KEGG enrichment analyses of the predicted targets revealed that “oxidoreductase activity” was among the top enriched terms in both yeast and human cells. These suggested a potential regulation of oxidative stress response (OSR) by NLE. RNA-seq analysis of NLE-treated yeast corroborated the anti-oxidative effect, with “oxidoreductase activity” and “oxidation-reduction process” ranking high in enriched GO terms. Notably, CTT1, encoding catalase, emerged as the most significantly up-regulated gene within the “oxidoreductase activity” cluster. In a *ctt1* null mutant, the enhanced oxidative stress resistance and extended lifespan induced by NLE were nullified. For human cells, NLE pretreatment demonstrated a decrease in reactive oxygen species levels and senescence-associated β-galactosidase activity in HeLa cells, indicative of anti-aging and anti-oxidative effects. This study unveils the anti-aging and anti-oxidative properties of NLE while delving into their mechanisms, providing novel insights for pharmacological interventions in aging using phytochemicals.

## 1. Introduction

*Azadirachta indica* (Neem), a medicinal plant of the mahogany family, has been utilized for centuries in India and Africa to treat both acute and chronic diseases. Preclinical studies have revealed that neem and its constituents are promising for the treatment of cancer [1,2], diabetes [3], skin diseases [4], and more [5,6]. Clinical trials have further shown the effectiveness of neem and its constituents against conditions like ulcer [7], psoriasis [8], and cervical cancer [9].

To date, over 300 chemicals have been identified in different parts of neem, targeting a wide range of proteins and cell signaling pathways [9]. Neem leaves, enriched with bioactive compounds such as glycoproteins, flavonoids, polyphenols, and isoprenoids, have been found to exert various effects on human cells. For instance, the glycoprotein found in neem leaves triggers the perforin-mediated destruction of tumor cells by T and NK cells via the selective modulation of IFNγ signaling [10]. Additionally, neem leaf glycoprotein restores dysregulated C-C motif chemokine signaling for monocyte/macrophage chemotaxis in patients with head and neck squamous cell carcinoma [11]. Neem leaf extract (NLE) has been shown to modulate pro-inflammatory pathways and reduce the proliferation, inflammation, migration, and invasion of human colon cancer cells [12]. Moreover, NLE significantly mitigated the H_2_O_2_-induced oxidative damage to red blood cells [13]. Neem leaves have traditionally been used for promoting longevity [14]. However, despite this traditional use, the anti-aging effects of neem leaves on human cells have scarcely been studied, and the potential mechanisms remain unaddressed.

The budding yeast, *Saccharomyces cerevisiae*, serves as a simple yet powerful eukaryotic model organism for aging research [15]. Yeast cells are easily grown and manipulated in the laboratory, and many aging-relevant pathways are highly conserved between yeast and humans, including nutrient and energy signaling, DNA repair, mitochondrial homeostasis, stress responses, and more [16]. Utilizing this unicellular system, potential anti-aging compounds can be screened more efficiently, and drug targets can be tested through yeast gene deletion or overexpression collections. Notably, yeast-based studies have previously led to the identification of potential anti-aging drugs such as resveratrol and spermidine [16]. Therefore, yeast can be also useful for understanding the potential anti-aging effects of neem leaf and its constituents.

Aging is a complex biological process involving numerous pathways and targets, and the compounds in neem leaf are diverse and complex. Thus, the traditional ‘one disease–one target–one drug’ dogma of medicine study is not suitable for comprehending the potential anti-aging effects of neem leaf. In this context, network pharmacology approaches that analyze the mechanistic actions of two or more drugs on multitargets become instrumental [17]. These approaches construct networks containing all the active compounds, targets and compound–target interactions [18]. The networks are constructed by firstly predicting the targets of the active compounds using one or more online platforms/databases. Then, the networks are constructed based on the topology of the active compounds, targets and compound–target interactions. Combined with Gene Ontology (GO) and Kyoto Encyclopedia of Genes and Genomes (KEGG) enrichment analysis, these networks provide better knowledge of the comprehensive effects of NLE and mechanisms underlying multitarget signaling modules. Due to the multi-compound and multi-target characteristics of photomedicine, network pharmacology approaches have been widely used in relevant studies [19,20,21]. However, network pharmacology studies of neem leaf extract have barely been undertaken.

A significant challenge for network pharmacology studies of anti-aging drugs is the validation of targets and phenotypes, a challenge partially addressed by employing the yeast system. In this study, we applied network pharmacology methods to identify aging-related pathways targeted by neem leaf compounds. These targets were further studied and validated in yeast. Our research contributes a mechanistic understanding of the anti-aging potential of neem leaves and serves as an example of anti-aging drug discovery through network pharmacology using yeast.

## 2. Materials and Methods

### 2.1. Strains and Cell Culture

The sources of yeast strains and human cells are described in Table 1. The SDC culture medium followed published procedures [22]. HeLa cervical cancer cells were cultured by Dulbecco’s Modified Eagle Medium (DMEM) with 4.5 g/L glucose (BasalMedia, Shanghai, China, L110KJ). Other reagents include serum (BasalMedia, S660KJ), Penicillin–Streptomycin solution (BasalMedia, S110JV) and pancreatin (BasalMedia, S310JV).

### 2.2. Preparation of Neem Leaf Extracts

Neem leaves, sourced from mountainous areas in the suburbs of Panzhihua City, Sichuan province, China, were dried to a constant weight and ground into a powder using a crusher, followed by sieving. The powdered material underwent ultrasound-assisted extraction [23] using a 50% or 75% ethanol–water solution (liquid to powder ratio of 12.5 mL/g) for 15 min. Subsequently, the extract was left to stand overnight at 4 °C and then centrifuged for 10 min at 20,000× *g*. The supernatant underwent an additional centrifugation step for 10 min at 20,000× *g* and was filtered through a 0.22 μm PES membrane.

### 2.3. Chronological Lifespan Assays

Chronological lifespan assays followed the published procedure [22]. In brief, yeast colonies were inoculated into 20 mL of YPD medium and incubated overnight at 30 °C. Subsequently, yeast cells were diluted into 20 mL of SDC medium (in a 50 mL flask) to achieve an initial A600 nm of 0.05. At the point of yeast cell inoculation, NLE or ethanol were added at the indicated concentration. The yeast cells in the SDC medium were then incubated at 30 °C in an air bath shaker (200 rpm) until reaching the stationary phase, and cell viability was assessed on day 0. This was undertaken by diluting the cells with 100 mM K-phosphate buffer (pH 6.0) and spreading them on YPD plates. The YPD plates were incubated at 30 °C for 72 h, after which colonies were counted. Then, at every indicated day X, cell viability was assessed. The survival rate was calculated by dividing the number of colonies at day X by the number of colonies at day 0.

### 2.4. Oxidative-Stress-Resistance Assays

Oxidative-stress-resistance assays followed the published procedure [24]. In brief, stationary cells were prepared in accordance with the protocol described for Chronological Lifespan assays. Subsequently, the cells were collected and resuspended in 1 mL of 100 mM K-phosphate buffer (pH 6.0) to achieve an A600 nm of 1. H_2_O_2_ was then added to a final concentration of 50 mM to induce oxidative damage. Following treatment with H_2_O_2_ for the specified duration at 30 °C, the cells were diluted 10, 10^2^, 10^3^ and 10^4^ times by 100 mM K-phosphate buffer (pH 6.0). Subsequently, 4.5 μL of each dilution was spread onto a YPD plate and incubated for 2 days at 30 °C to test viability. The results were photographed and then the pictures were converted into black and white format by PowerPoint software(2404 Build 16.0.17531.20140 64-bit).

### 2.5. Reactive Oxygen Species Measurement

To measure reactive oxygen species (ROS) in yeast cells [25], stationary cells were prepared following the procedure outlined in the Chronological Lifespan assay. The cells were then collected and resuspended in 1 mL of PBS buffer to achieve an A600 nm of 1. The cell suspension was incubated with 10 μM DCFH-DA at 30 °C for 60 min. Following staining, the cells were thoroughly washed twice with PBS buffer and subsequently re-suspended in 1 mL of PBS. DCFH-DA-stained cells were examined using fluorescence microscopy (excitation/emission: 488 nm/530 nm). For ROS measurements in HeLa cells, approximately 5 × 10^5^ cells/well were seeded into a 6-well plate. After 12 h of incubation, NLE or 50% ethanol was added to a final concentration of 0.3%. Following a 12 h treatment, cells were washed twice with PBS, and 1 mL of 10 μM DCFH-DA (in PBS)/well was added. After 30 min of staining, cells were washed twice with PBS and 1 mL of 1 mM H_2_O_2_ (in PBS)/well was added. After an additional 30 min, cells were washed twice with PBS and observed using fluorescence microscopy (excitation/emission: 488 nm/530 nm). The fluorescence intensity of the DCFH-DA-stained cells was quantified by ImageJ software (1.48v) as the value of integrated density (IntDen).

### 2.6. Analysis of Active Compounds of NLE

Initially, LC-MS (EKspert nanoLC415/ Triple TOF 5600, SCIEX, Singapore) was employed to determine the constituents of 50% ethanol NLE. The LC-MS analytical conditions were as follows: the samples were separated using an Agilent ZORBAX SB-C18 Rapid Resolution HD (Agilent, Santa Clara, CA, USA) 2.1 × 100 mm 1.8-micron column, and the column temperature was maintained at 40 °C. The flow rate was set at 0.4 mL/min. The mobile phases consisted of 0.1% formic acid (A) and acetonitrile (B). The gradient was programmed as follows: 5% B for 0.5 min; 5–95% B over 11 min; and 95% B for 2.5 min, followed by a decrease to 5% B for 0.1 min and re-equilibration of the column for 2.4 min with 5% B. The injection volume was 1 μL, and the sampler temperature was set at 15 °C. Mass spectrometry employed an electrospray ionization (ESI) source. The source parameters in positive/negative polarity were set as follows: ion source gas 1:50 psi; ion source gas 2:50 psi; curtain gas: 35 psi; CAD gas: medium; temperature: 500 °C; and spray voltage: 5500/−4500 V.

The LC-MS results were transformed to mzML format using Proteowizard software (3.0.23199 64-bit) with the ‘Peak Picking’ filter (parameters: vendor msLevel = 1-). Subsequently, the result files were uploaded to Global Natural Products Social Molecular Networking (GNPS, https://gnps.ucsd.edu/, accessed on 20 July 2023) for molecular network analysis. The outcomes of the molecular networking analysis can be accessed on two websites: (https://gnps.ucsd.edu/ProteoSAFe/status.jsp?task=14b9aa1c8e474f80a5ad474d3782d8fb (accessed on 21 July 2023) and https://gnps.ucsd.edu/ProteoSAFe/status.jsp?task=e3bac4a72ff243529d67b8c1082928ed (accessed on 21 July 2023)). The parameter settings are available on these websites as well. In summary, Precursor Ion Mass Tolerance and Fragment Ion Mass Tolerance were set to 0.01 and 0.02, respectively; other parameters were left at their default values. Compounds identified through molecular networking were filtered based on data information, including ‘wrong IonMode’, ‘N/A InChiKey’, and ‘TIC_Query < 2000’. The remaining compounds with high-quality data were identified as potential active compounds of NLE, as shown in Appendix A. Their names were retrieved from PubChem (https://pubchem.ncbi.nlm.nih.gov/, accessed on 24 July 2023) based on their InChiKey.

### 2.7. Target Prediction of NLE

To predict the potential targets of the identified potential active compounds of NLE, the InChiKey or SMILES of these compounds were uploaded to the Swiss Target Prediction platform (STP, http://www.swisstargetprediction.ch/index.php/, accessed on 24 July 2023) and STITCH platform (https://stitch.embl.de//, accessed on 24 July 2023). Gene Ontology (GO) and Kyoto encyclopedia of genes and genomes (KEGG) enrichment analysis of the targets were performed using the Metascape platform (www.metascape.org//, accessed on 30 July 2023).

### 2.8. RNA-Seq

The preparation of stationary cells adhered to the protocol outlined in the Chronological Lifespan assay. Four biological replicates were collected and promptly snap-frozen in liquid nitrogen for subsequent total RNA extraction. The mRNA samples were sequenced on an Illumina HiSeq 2500 platform (Illumina, San Diego, CA, USA), and paired-end 150-bp reads were generated following the published procedure [26]. Reads containing greater than 10% poly-N and greater than 50% low-quality reads (Q ≤ 20) were removed from the raw data using Trimmomatic v0.33. Simultaneously, Q20 and Q30 values, GC content, and sequence duplication levels of the clean data were computed. All subsequent analyses were conducted using clean and high-quality data. The RNA-Seq reads underwent alignment to the yeast reference genome (Saccharomyces cerevisiae, Ensembl_R64-1-1) employing TopHat 2 (v2.1.0). Each read was mapped using Cufflinks (v2.1.1), a program that assembled the alignments in the Sequence Alignment/Map format into transfrags. Subsequently, the assembly files were merged with reference transcriptome annotations for further analysis. The differential expression of each gene was computed by quantifying the Illumina reads based on fragments per kilobase of transcript per million fragments mapped (FPKM). Gene Ontology (GO) and Kyoto Encyclopedia of Genes and Genomes (KEGG) enrichment analysis for the Differentially Expressed Genes (DEGs) were carried out using the Metascape platform (www.metascape.org/).

### 2.9. RT-qPCR

Total RNA extraction was carried out using Beyozol Reagent in accordance with the manufacturer’s instructions (Beyotime, Shanghai, China, R0011). Subsequently, genomic DNA was eliminated, and cDNA was synthesized through reverse transcription using the BeyoRT™ II First Strand cDNA Synthesis Kit with gDNA Eraser (Beyotime, D7170M). The primers utilized in this study are detailed in Table 2, and the HotStart™ Universal 2X SYBR Green qPCR Master Mix (APExBIO, Houston, TX, USA, K1170) was employed. For the RT-qPCR steps, the settings were configured as follows: 95 °C for 2 min (1 cycle); 95 °C for 15 s, 60 °C for 30 s (40 cycles); and 95 °C for 15 s, 60 °C for 60 s, 95 °C for 15 s (1 cycle).

### 2.10. Senescence-Associated β-Galactosidase Staining

HeLa cells were seeded at 1 × 10^5^ cells/well on 6-well plates. After 12 h of culture, NLE or 50% ethanol was added at a final concentration of 0.3%. After 12 h of treatment, these cells were stained using a Senescence β-Galactosidase Staining Kit (Beyotime, C0602).

### 2.11. Statistical Analysis

The data are expressed as mean ± S.E.M. Statistical comparisons were performed using Student’s *t*-test. * *p* < 0.05, ** *p* < 0.01, *** *p* < 0.001, **** *p* < 0.0001.

## 3. Results

### 3.1. NLE Extends the Lifespan of Yeast Cells

To assess the potential anti-aging effects of neem, we initially obtained neem leaf extracts (NLE) using varying concentrations of ethanol solutions. By testing the chronological lifespan of the yeast strain BY4742 in the presence of NLEs, we observed that NLE extracted from 50% or 75% ethanol significantly extended the cells’ lifespans, as depicted in Figure 1A,B. Consequently, in subsequent studies, we utilized NLE extracted from 50% ethanol to explore its anti-aging effects.

BY4742 is an engineered haploid MATalpha strain derived from the S288C background, sharing similarities with another commonly used laboratory MATa strain, BY4741 (PMID:9483801). While the variations between BY4742 and BY4742 are minimal, differences in selectable marker genes and mating types may influence their lifespan response to nutritional and environmental signals. Therefore, we investigated whether NLE also extends the lifespan of BY4741 cells. Our findings revealed that 0.2% of NLE slightly, yet significantly, extended the lifespan of BY4741 cells, as illustrated in Figure 1C. Furthermore, increasing the concentration of NLE to 0.8% resulted in additional lifespan extension, although not to the extent observed in BY4742 cells. This suggests that, even though BY4741 and BY4742 cells exhibit differential responses to NLE, NLE effectively extends the lifespan of both strains.

### 3.2. NLE Enhances the Oxidative Stress Response (OSR) in Yeast Cells

In general, the extension of the chronological lifespan of yeast cells is closely linked to an improved oxidative stress response (OSR). To investigate the mechanism by which NLE extends lifespan, we initially examined the impact of NLE on OSR. When BY4742 cells were exposed to 50 mM H_2_O_2_ in YPD medium for 60 or 120 min, a majority of the cells died. However, the pre-incubation of cells with NLE significantly promoted survival under H_2_O_2_-induced oxidative stress (Figure 2A). A similar effect of NLE on OSR in BY4741 was also observed (Figure 2B).

Using fluorescent ROS probe DCFH-DA, we monitored the relative level of ROS in yeast cells when exogenous H_2_O_2_ is added into culture medium. The addition of exogenous H_2_O_2_ significantly increased the level of intracellular ROS, which was attenuated by the pre-incubation of NLE (Figure 2C). These results suggest that NLE has the ability to suppress oxidative stress in yeast cells.

### 3.3. Active Compounds and Target Prediction of NLE

The effects of NLE on lifespan and the OSR of yeast cells raise the question of which compounds in NLE are responsible for those effects. To address this question, the metabolites with NLE extracted from 50% ethanol were analyzed by liquid chromatography–tandem mass spectrometry (LC-MS/MS). The resulting spectra were submitted to the GNPS platform, identifying 80 compounds (Appendix A), with flavonoids being the predominant constituents. Details of the compounds are shown in Appendix A.

To estimate the potential effects of NLE compounds on yeast and human cells, we utilized both the STITCH platform and the Swiss Target Prediction (STP) platform to predict the potential targets of these compounds. For yeast, STITCH identified a total of 378 potential gene targets for 28 compounds. The top six compounds with the most targets were quercetin, kaempferol, phloretin, nonaethylene glycol, trehalose, and L-epicatechin (223, 110, 54, 44, and 43 targets, respectively). The top seven targets with the most compounds were ERG5, DIT2, YPK2, YPK3, YPK1, SCH9, and HMX1, each associated with seven compounds. The compound–target network details are presented in Figure 3A. To investigate the critical molecular functions and pathways of all these predicted targets, GOMF and KEGG enrichment was conducted. The results indicate that “oxidoreductase activity” (Figure 3B) and “longevity-regulating pathway” (Figure 3C) are among the top 15 enriched terms, supporting the observed anti-oxidative and lifespan-extension effects of NLE (Figure 1 and Figure 2).

For human cells, a total of 621 potential gene targets for 53 compounds were predicted by both STITCH and STP. The top six compounds with the most targets were quercetin, kaempferol, myricetin, trehalose, isorhamnetin, and nonaethylene glycol (363, 240, 139, 95, 90, and 76 targets, respectively). The top seven targets with the most compounds were AKR1B1, CA7, CA12, ACHE, CA2, NOX4, and CA4 (24, 21, 21, 19, 19, 18, and 17 compounds, respectively). Details of the compound–target network are shown in Figure 3D. To investigate the critical molecular functions and pathways of all these predicted targets, GOMF and KEGG enrichment was conducted. The results show that “oxidoreductase activity” (Figure 3E) and “chemical carcinogenesis—reactive oxygen species” (Figure 3F) are two of the top 15 enriched terms, suggesting that NLE may regulate the OSR of human cells.

### 3.4. NLE Induces Transcriptome Changes to OSR in Yeast

To elucidate the mechanism by which NLE regulates OSR and lifespan, we conducted a transcriptome analysis to examine changes in the gene expression profile of BY4742 cells treated with NLE. A total of 6031 unigenes were expressed in both control and NLE-treated yeasts, with 31 unigenes expressed exclusively in control yeasts and 37 unigenes expressed solely in NLE-treated yeasts (Figure 4A). In total, 1092 differentially expressed genes (DEGs) were identified, comprising 648 upregulated genes and 444 downregulated genes (Figure 4B).

To investigate critical biological processes and molecular functions, we performed GO enrichment analysis on these 1092 DEGs using GOseq [27]. GO analysis demonstrates that “oxidoreductase activity” is one of the top 15 GOMF terms (Figure 4C) and “oxidation-reduction process” is one of the top 15 GOBP terms (Figure 4D), confirming that NLE regulates the OSR of yeasts. The top 10 upregulated genes of “oxidoreductase activity” are CTT1, TKL2, COX4, QCR8, GND2, MRH1, GRE2, MXR1, HFD1, and GRX4 (Figure 4E), and the top 10 upregulated genes of the “oxidation-reduction process” are CTT1, RGI1, GSY2, GPH1, GND2, MRH1, QCR9, BDH1, GRE2, and MXR1 (Figure 4F).

### 3.5. Oxidoreductases Are Involved in Enhanced OSR Induced by NLE

As NLE enhances OSR, we hypothesize that differentially expressed genes (DEGs) enriched in the Gene Ontology terms “oxidation-reduction process” and “oxidoreductase activity” may play crucial roles in the antioxidative and antiaging effects of NLE. Firstly, we sought to determine whether the upregulation effects of NLE on oxidoreductases are unique to the BY4742 strain. To address this, the six most significant DEGs in these GO terms were chosen, including CTT1, TKL2, COX4, QCR8, GND2, and QCR9, and their expression in NLE-treated BY4741 cells was monitored by using qPCR. Similar to the results obtained by the transcriptome sequencing of NLE-treated BY4742, the expression of these genes is also significantly upregulated in NLE-treated BY4741 cells (Figure 5A). Notably, CTT1, which encodes catalase in yeast, is the most significantly up-regulated oxidoreductase gene in both BY4742 and BY4741.

Next, we asked if these oxidoreductases are required for the anti-oxidative effects of NLE. *cct1*, *qcr8*, *gnd2*, and *qcr9* null mutants of BY4741 background were obtained from the *Saccharomyces cerevisiae* Genome Deletion Project and their OSR against H_2_O_2_ with or without NLE treatment was tested. When cells of the stationary growth stage were treated with 50 mM of H_2_O_2_, *ctt1* mutant pre-treated with or without NLE was not viable (Figure 5B). *qcr8*, *gnd2*, and *qcr9* mutants were more sensitive to H_2_O_2_ than the wild type, and NLE could enhance their OSR (Figure 5B), indicating that these three genes are related to OSR but not necessary for the effects of NLE.

Given the extreme sensitivity of the *ctt1* mutant to 50 mM H_2_O_2_, we lowered the concentration of H_2_O_2_ to 5 mM to evaluate the role of CTT1 for the effects of NLE. The *ctt1* mutant is viable for 120 min in 5 mM of H_2_O_2_. Interestingly, the pre-treatment of NLE had no protective effects on the *ctt1* mutant (Figure 5C). These results indicate that CTT1 is the key factor in counteracting H_2_O_2_-induced oxidative stress, and the protective effect of NLE against H_2_O_2_ also relies on it.

### 3.6. CCT1 Is Required for the Lifespan-Extension Effects of NLE

Because CTT1 is necessary for NLE to enhance yeast cell resistance against H_2_O_2_-induced oxidative stress, we investigated whether CTT1 also plays a crucial role in the lifespan extension induced by NLE. Comparing the lifespan of the *ctt1* mutant to wild-type BY4741 cells revealed that the *ctt1* mutant lived longer than the wild-type cells (Figure 6). However, while NLE extends the lifespan of wild-type cells, it fails to extend the lifespan of the *ctt1* mutant under the same conditions. These results suggest that, although the absence of CTT1 does not shorten the lifespan, the lifespan extension effects of NLE require CTT1.

Previous experiments have demonstrated that NLE could decrease the level of intracellular ROS of wild-type yeasts (Figure 2C,D), and CTT1 plays an important role in NLE-induced oxidative stress resistance promotion (Figure 5C). We then aimed to determine whether CTT1 also plays an important role in NLE-induced ROS reduction. However, when the *ctt1* null mutant was treated with NLE, the level of intracellular ROS still significantly decreased (Figure 6C), indicating that NLE-induced ROS reduction does not solely depend on CTT1. In addition to CTT1, there are other factors involved in decreasing the ROS level of yeast cells after NLE treatment.

To explore the regulatory dynamics of the NLE compound–target network on CTT1, we formulated a comprehensive interaction network, encompassing CTT1, relevant compounds, and targets. This network delineates both direct and indirect connections, providing insight into the intricate relationships between CTT1 and its associated compounds and targets (Figure 6D). A total of 8 compounds and 31 targets from the compound–target network were identified. The results show that the top five targets with the highest degree values were HSP104, NTH1, TPS2, HXK1, and PGM2 (18, 16, 16, 16, and 16, respectively). Meanwhile, the top two compounds with the highest degree value were trehalose and quercetin (15 and 10, respectively). Additionally, the top target with the most compounds was HOG1, which interacts with kaempferol, procyanidin B2, N-oleoylethanolamine, and quercetin. Among these 31 targets, HOG1, MSN2, and MSN4 are transcription factors of CTT1.

### 3.7. NLE Promotes Resistance against H_2_O_2_-Induced Oxidative Stress in Human Cells

Since NLE enhanced OSR in yeast cells (Figure 2) and was predicted to regulate OSR in human cells (Figure 3E,F), we then asked whether NLE is able to promote resistance to H_2_O_2_-induced oxidative stress and the senescence of human cells. Firstly, we assessed the ROS of HeLa cells treated with or without NLE. Under H_2_O_2_-induced oxidative stress, HeLa cells treated with NLE produced fewer ROS compared to the control (Figure 7A). This result indicates that NLE enhances the OSR of human cells. We then evaluated the H_2_O_2_-induced senescence of HeLa cells through senescence-associated β-galactosidase (SA-β-gal) staining. HeLa cells treated with H_2_O_2_ exhibited strong SA-β-gal staining, indicating the effective induction of senescence by H_2_O_2_ (Figure 7B). NLE pre-treatment significantly attenuates H_2_O_2_-induced SA-β-gal activation, indicating that NLE treatment also prevents cell senescence induced by oxidative stress in human cells. These data suggest that the antioxidative and anti-senescence/anti-aging effects of NLE are conserved from yeast to human cells.

## 4. Discussion

Pharmacological intervention aimed at extending lifespan represents a promising strategy, with oxidative damage prevention emerging as a pivotal factor in achieving this goal [28]. In our study, we identified both lifespan extension and anti-oxidative effects associated with neem leaf extract (NLE) in yeast. The straightforward preparation method for NLE, involving crushing, soaking, and ultrasound treatment without the need for further purification [14], adds practicality and convenience to its utilization. These attributes enhance the potential application of NLE, making it a feasible and promising candidate as a tonic or food additive to mitigate the aging process.

To enhance the practical application of NLE, a comprehensive understanding of its constituents and the underlying mechanisms responsible for its anti-oxidative and lifespan-extension effects is essential. In this pursuit, we employed LC/MS and conducted molecular network analyses on the GNPS platform to identify potential active compounds within NLE. Our analysis revealed the presence of various natural chemicals, including quercetin, kaempferol, phloretin, nonaethylene glycol, trehalose, and L-epicatechin, among others. A significant portion of these compounds belong to the flavonoid class, which has previously been reported for its anti-oxidative properties [29,30,31,32]. This finding provides a plausible explanation for the observed anti-oxidative effects of NLE.

Aging is a multifaceted biological process, and the active compounds found in NLE target multiple pathways. Therefore, network pharmacological analyses were conducted to explore the anti-aging and antioxidant effects of NLE and their underlying mechanisms. Initially, potential targets of each compound were identified using the STP and STITCH platforms, which generate targets based on experimental results or theoretical computation. Since results may vary between platforms, both were utilized to ensure comprehensive findings. Subsequently, pharmacological networks were constructed based on the topology of the active compounds, targets, and compound–target interactions. From these networks, targets with relatively high degrees were identified, indicating their interaction with a significant number of compounds. However, relying solely on high-degree targets posed limitations, as a single target might not be reliable due to unknown compound doses and high-degree targets might not fully represent the comprehensive effects of all targets. To mitigate errors associated with single targets and better understand the overall effects of NLE, GO and KEGG enrichment analyses were performed. Interestingly, the results revealed that “oxidoreductase activity” and “longevity-regulating pathways” were among the top enriched terms, supporting the observed antioxidative and lifespan-extension effects of NLE in yeast cells. Similarly, in human cells, the analysis indicated that “oxidoreductase activity” and “chemical carcinogenesis—reactive oxygen species” were among the top enriched terms, corroborating the antioxidant effect of NLE on HeLa cells.

In alignment with network pharmacological analyses, our transcriptome investigation revealed the regulatory influence of NLE in yeast on “oxidoreductase activity” and “oxidation-reduction processes”. Further experimental studies indicated that catalase CTT1 prominently facilitated the NLE-induced extension of lifespan. Intriguingly, molecular network analyses did not identify CTT1 as the primary target with the highest degree value. Therefore, although network pharmacological analyses of NLE compounds offer valuable insights into targeted proteins and pathways, a comprehensive understanding of the underlying mechanism necessitates further experimental analysis.

The preceding findings prompt the question of why CTT1 experienced upregulation. To address this, we probed whether NLE targeted the regulators of CTT1. Among the array of predicted NLE targets, we focused on the manually curated regulators of CTT1, namely HOG1, MSN2, and MSN4, as documented in the Saccharomyces Genome Database (https://www.yeastgenome.org/, accessed on 8 August 2023). These three regulators govern the transcription of CTT1, with degrees in the pharmacological networks measuring 5, 1, and 1, respectively. Notably, HOG1 exhibits a higher degree, suggesting its potential role in orchestrating the upregulation of CTT1 under NLE treatment. Despite this insight, the underlying cause for the NLE-induced upregulation of CTT1 remains elusive based on current data and the published literature. Further experiments are imperative to scrutinize alterations in the regulators of CTT1 under NLE treatment and unravel the mechanisms at play.

Considering evolutionary conservation and pivotal role of the OSR pathway in cellular aging across yeast and human systems, we further investigated whether NLE could safeguard human cells against H_2_O_2_-induced oxidative stress and aging. The measurement of ROS levels and SA-β-gal staining in HeLa cells revealed that pre-treatment with NLE significantly reduced intracellular ROS levels and mitigated H_2_O_2_-induced SA-β-gal activation. These findings suggest that NLE enhances oxidative stress resistance and alleviates oxidative-stress-induced senescence in human cells. Notably, CTT1 has a human ortholog called CAT, a crucial antioxidant enzyme defending the body against oxidative stress. While certain compounds in NLE, such as quercetin [33] and myricetin [34], are reported to increase CAT activity, future studies are required to ascertain whether NLE upregulates CAT, thereby protecting various human cells from oxidative stress.

## 5. Conclusions

In this investigation, we prepared a neem leaf extract (NLE) and discerned its noteworthy anti-oxidative and anti-aging effects in both yeast and human cells. Through the application of LC/MS, we successfully identified the active compounds within NLE. Employing pharmacological network analysis, we predicted potential targets of NLE in yeast, highlighting oxidoreductase activity and the regulation of longevity pathways. Subsequent RNA-seq experiments confirmed the regulatory influence of NLE on oxidoreductase activity in yeast. Notably, we pinpointed CTT1 as the primary gene target of NLE in yeast and observed NLE’s potent transcriptional activation of CTT1 in yeast. Future studies will concentrate on investigating the antioxidative and anti-aging effects of NLE in animal models, as well as exploring its potential clinical applications.

## Figures and Tables

**Figure 1 nutrients-16-01506-f001:**
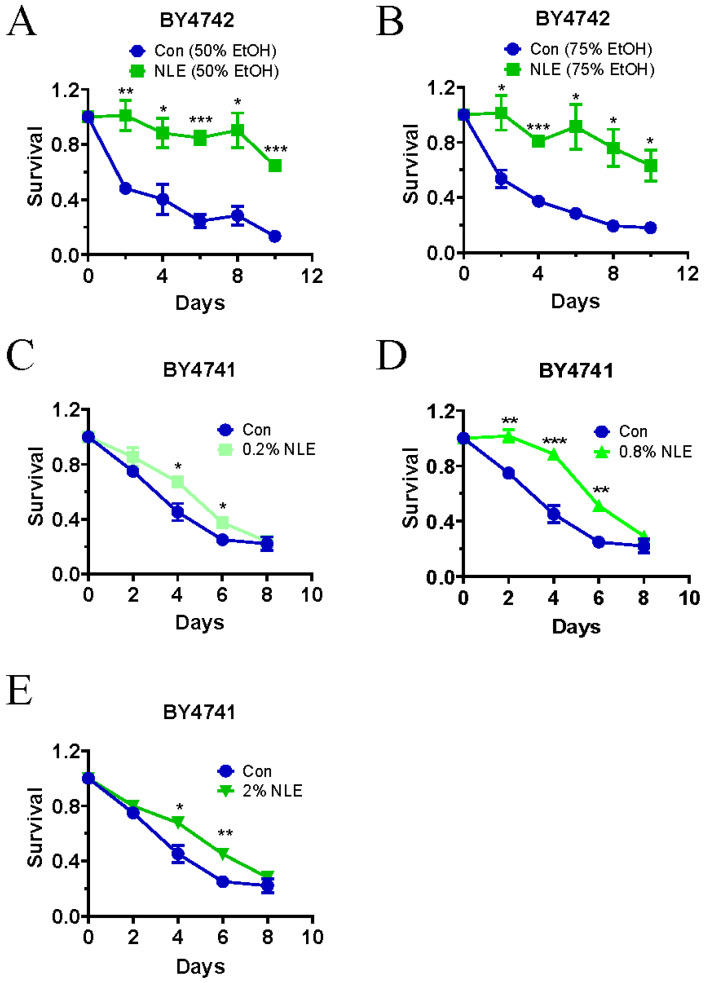
NLE extends the chronological lifespan of yeast. (**A**,**B**) BY4742 yeast cells were inoculated at an initial OD_600 nm_ of 0.05 and treated with either 50% ethanol NLE (**A**) or 75% ethanol NLE (**B**) at a final concentration of 0.5%, or left untreated. After growing into stationary phase, the chronological lifespan was measured. Triplicate cultures were used to achieve the mean ± SEM of survival rates (*n* = 3, Student’s *t*-test for each point, * *p* < 0.05, ** *p* < 0.01, *** *p* < 0.001). (**C**–**E**) BY4741 yeast cells were treated with or without 50% ethanol NLE with a final concentration of 0.2% (**C**), 0.8% (**D**), and 2% (**E**) while inoculated with the initial OD_600 nm_ of 0.05. After growing into the stationary phase, the chronological lifespan was measured. Quadruplicate cultures were used to achieve mean ± SEM of survival rates (*n* = 4, Student’s *t*-test for each point, * *p* < 0.05, ** *p* < 0.01, *** *p* < 0.001).

**Figure 2 nutrients-16-01506-f002:**
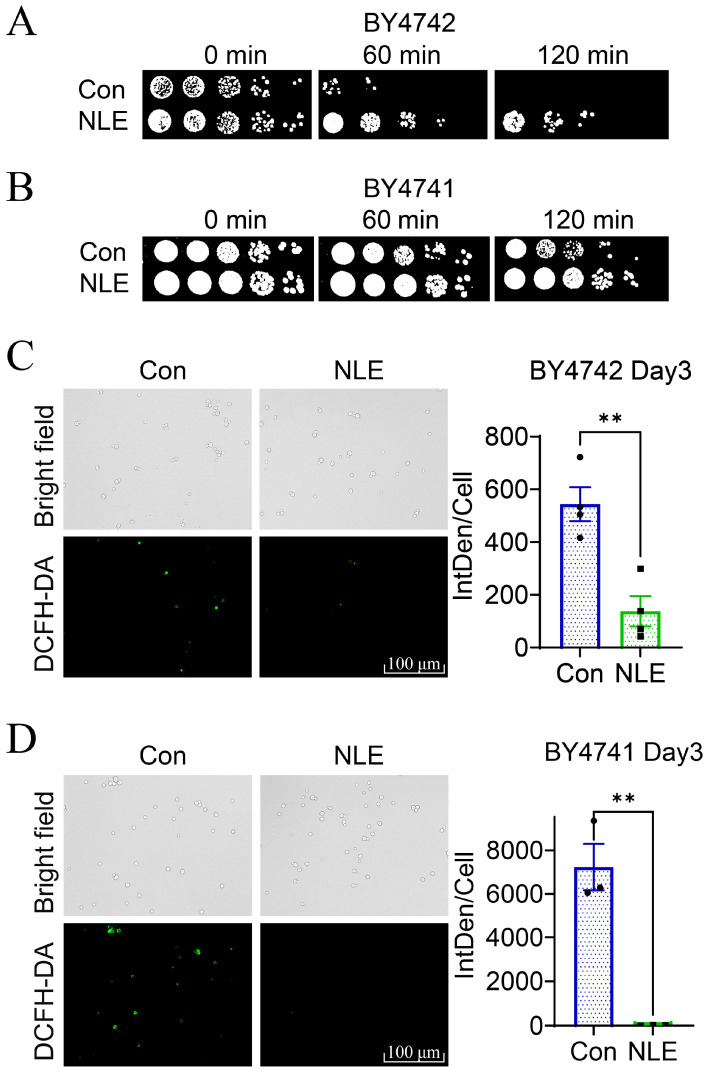
NLE enhances the OSR of yeast. (**A**,**B**) Resistance of NLE-treated BY4742 (**A**) and BY4741 (**B**) cells to 50 mM hydrogen peroxide (H_2_O_2_) stress at stationary phase. Photographs show the colonies of a ten-fold dilution series of yeast cells (from (**left**) to (**right**)). (**C**,**D**) The fluorescence of DCFH-DA, which indicates the generation of reactive oxygen species (ROS) in BY4742 (**C**) and BY4741 (**D**) during the stationary phase under NLE treatment, was captured using a fluorescence microscope (**left**) and quantified by calculating the integrated density per cell (IntDen/Cell) of positively stained cells (**right**). *n* = 4 (**C**) or 3 (**D**), Student’s *t*-test, ** *p* < 0.01.

**Figure 3 nutrients-16-01506-f003:**
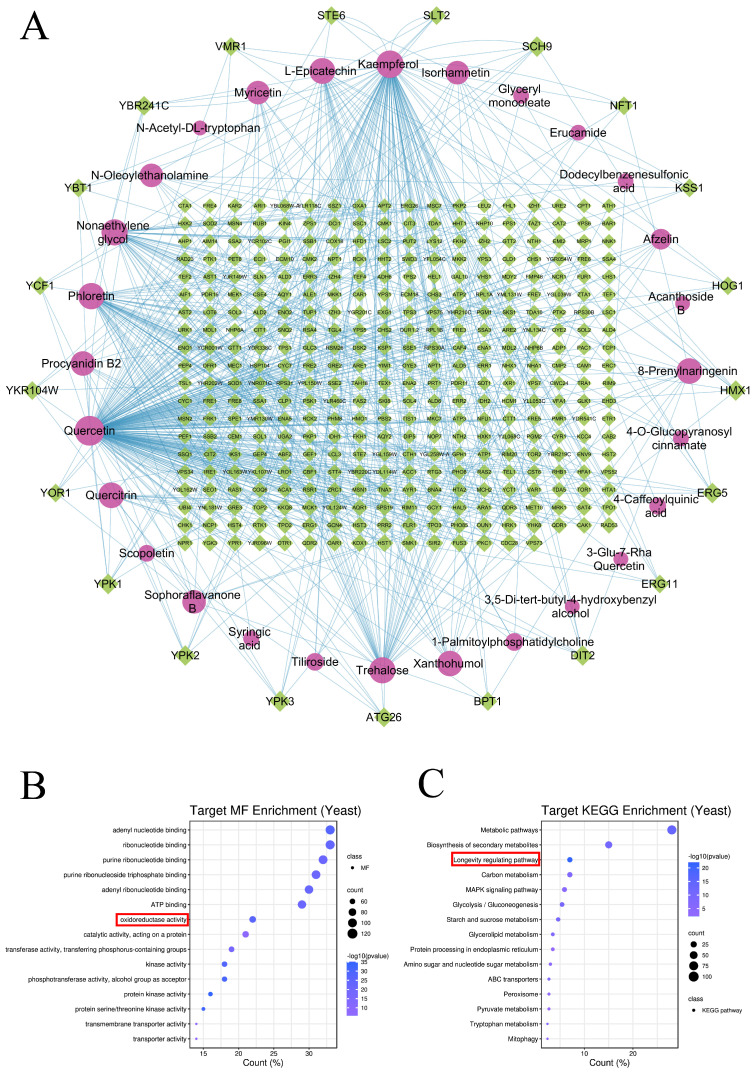
Network pharmacological analyses indicate that NLE may target the oxidative stress response of both yeast and human cells. (**A**) The network for compound–target connection in yeast cells. The ellipse nodes represent the compounds and diamond nodes represent targets. All nodes’ area changes are shown according to their degree value, and only gene nodes whose degree was greater than 4 are presented in the outer circle for better clarity. Lines between two nodes represent the predicted interaction. (**B**,**C**) The dot plot shows the enrichment results of molecular function (**B**) and KEGG (**C**) from total targets based on the network of yeast cells. (**D**) The network for compound–target connection in human cells. (**E**,**F**) The dot plot shows the enrichment results of molecular function (**E**) and KEGG (**F**) from total targets based on the network of human cells. Terms related to oxidative stress response (OSR) are marked by red boxes.

**Figure 4 nutrients-16-01506-f004:**
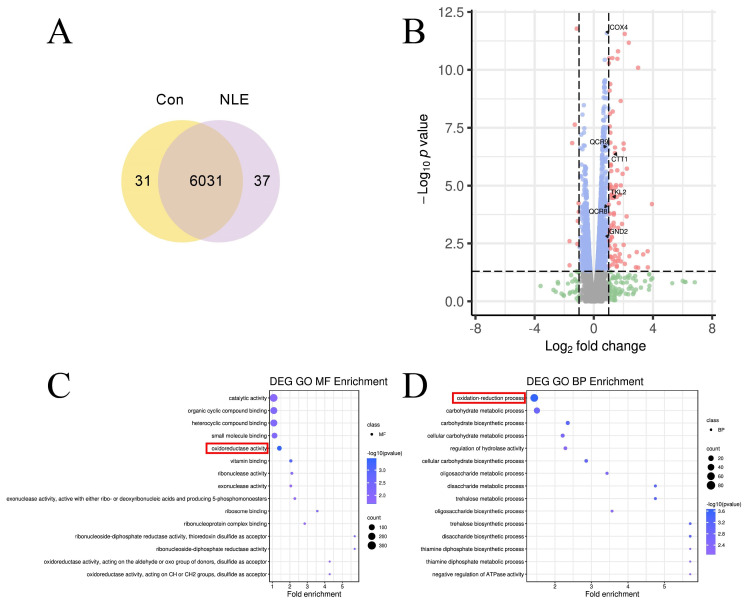
The transcriptome of NLE-treated yeast. (**A**) The Venn plot of unigenes. (**B**) The volcano plot of gene expression in NLE-treated BY4742 cells compared to untreated cells. The top DEGs of oxidoreductases are indicated with arrows. Grey: Not significant, Green: Significant only in fold change, Blue: Significant only in *p* value, Pink: Significant in both fold change and *p* value. (**C**,**D**) The dot plot shows the enrichment results of molecular function (**C**) and biological process (**D**). The OSR-related terms are indicated with red boxes. (**E**,**F**) The heat maps of the top 10 upregulated DEGs of oxidoreductase activity (**E**) and the oxidation-reduction process (**F**).

**Figure 5 nutrients-16-01506-f005:**
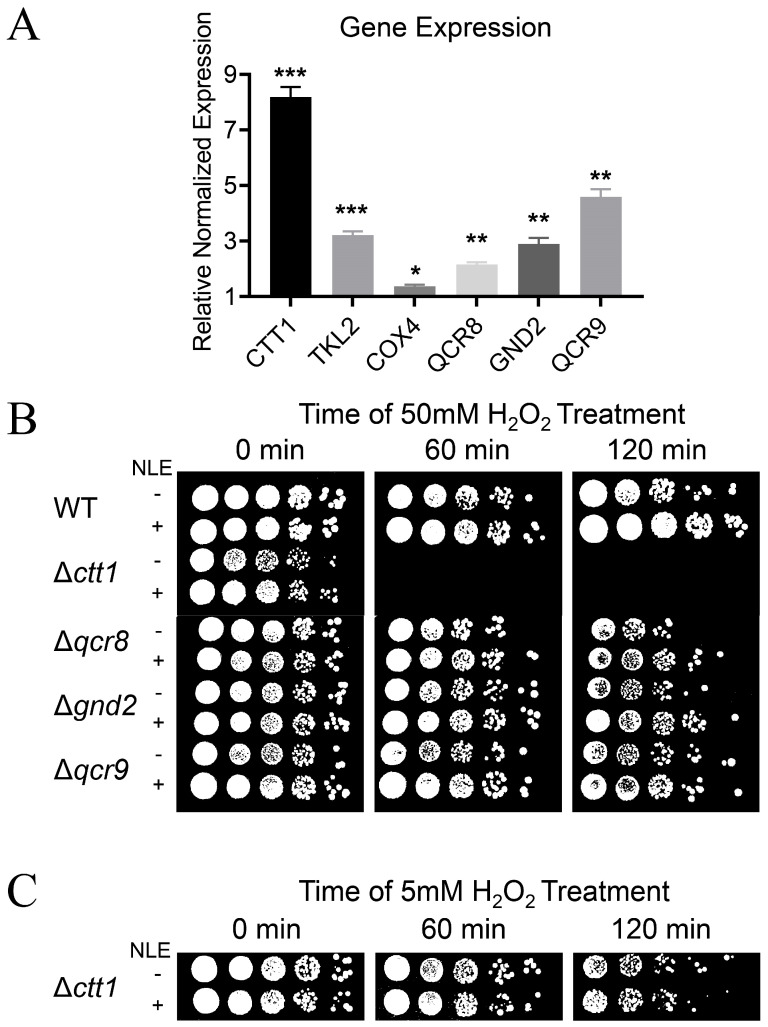
Oxidoreductases are involved in enhanced OSR induced by NLE. (**A**) RT-qPCR results of the top changed oxidoreductases of NLE-treated BY4741 (*n* = 3, * *p* < 0.05, ** *p* < 0.01, *** *p* < 0.001). (**B**) Resistance of NLE-treated null mutants with BY4741 background against 50 mM H_2_O_2_. (**C**) Resistance of NLE-treated Δ*ctt1* null mutants with BY4741 background against 50 mM H_2_O_2_.

**Figure 6 nutrients-16-01506-f006:**
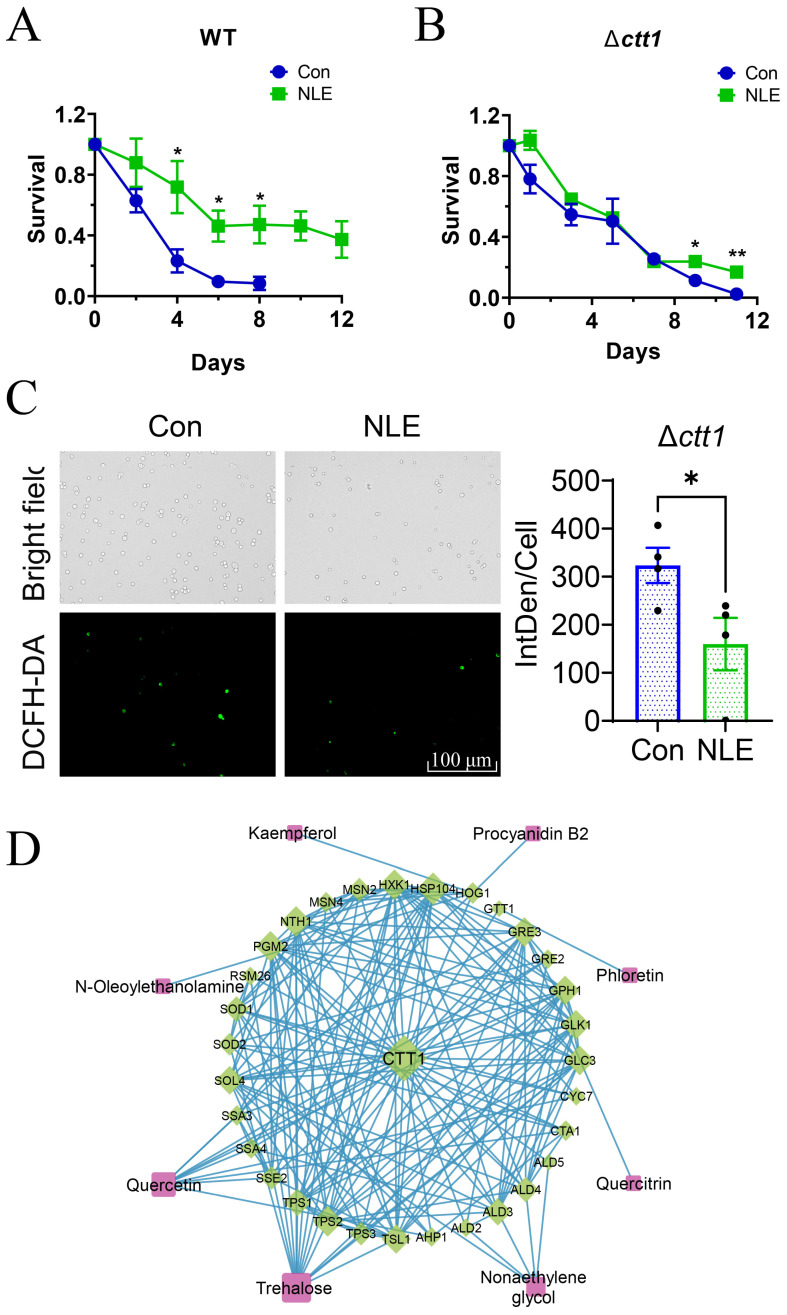
The roles of CTT1 in NLE-treated yeast cells. (**A**,**B**) The chronological lifespan of NLE-treated BY4741 wild-type strain (**A**) and its Δ*ctt1* null mutant (**B**) (*n* = 4, Student’s *t*-test for each point, * *p* < 0.05, ** *p* < 0.01). (**C**) The fluorescence of DCFH-DA, indicating the generation of ROS in Δ*ctt1* during the stationary phase under NLE treatment, was captured using a fluorescence microscope (**left**) and quantified by calculating the integrated density per cell (IntDen/Cell) of positively stained cells (**right**). (**D**) The network of the CTT1 target compound. The diamond nodes represent gene targets and round rectangle nodes represent compounds.

**Figure 7 nutrients-16-01506-f007:**
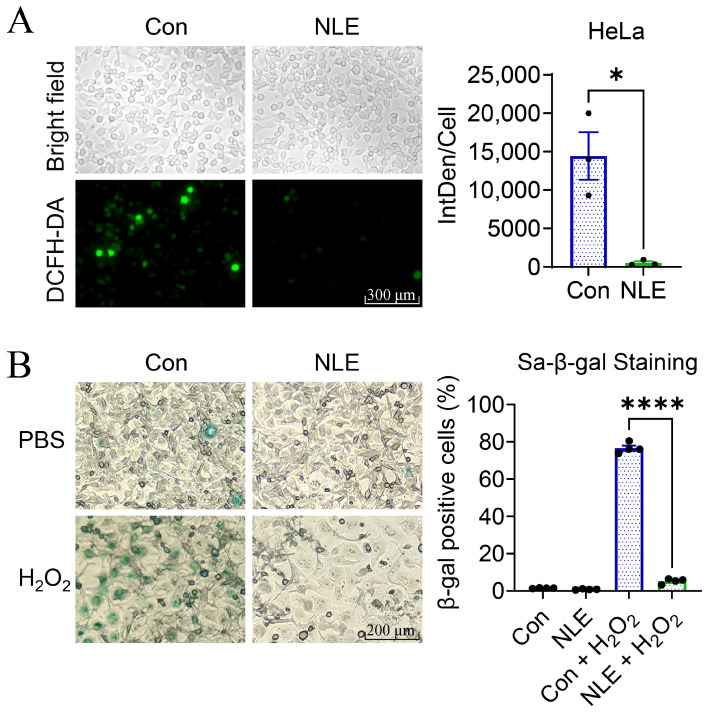
NLE promotes the resistance of HeLa cells against H_2_O_2_-induced oxidative stress. (**A**) The fluorescence of DCFH-DA, indicating the generation of ROS in HeLa cells, was visualized using a fluorescence microscope (**left**) and quantified by calculating the integrated density per cell (IntDen/Cell) of positively stained cells (**right**). HeLa cells were pretreated with or without NLE, followed by exposure to H_2_O_2_ to induce ROS generation. (*n* = 3, * *p* < 0.05). (**B**) The SA-β-gal staining of HeLa cells was imaged by microscope (**left**) and quantified by calculating the positively stained cells (**right**). HeLa cells were pretreated with or without NLE then exposed to PBS as control or to H_2_O_2_ to induce senescence (*n* = 4, **** *p* < 0.0001).

**Table 1 nutrients-16-01506-t001:** Strains and cells used in this research.

Strain/Cell Line	Genotype	Source
BY4741	MATa his3-∆1 leu2-∆0 ura3-∆0 met15-∆0	Lab stock
BY4742	MATalpha his3-∆1 leu2-∆0 ura3-∆0 lys2-∆0	Lab stock
Δ*ctt1*	BY4741 with ctt1::KAN	Saccharomyces Genome Deletion Project
Δ*qcr8*	BY4741 with qcr8::KAN	Saccharomyces Genome Deletion Project
Δ*gnd2*	BY4741 with gnd2::KAN	Saccharomyces Genome Deletion Project
Δ*qcr9*	BY4741 with qcr9::KAN	Saccharomyces Genome Deletion Project
HeLa		ATCC

**Table 2 nutrients-16-01506-t002:** Primers used in this research.

Gene	Primer (5′ to 3′)	
	Forward	Reverse
Actin	CCATCCAAGCCGTTTTGTCC	TGAGCAGCGGTTTGCATTTC
CTT1	CATGCCAAAGGTGGTGGTTG	CAGTCATGGTTCCCCCACTC
TKL2	GGTGTGAGGGAACACGGAAT	TTGGACCATCCTCACCAAGC
COX4	TTCAGCAAAAACCCGTGGTG	AGAACCCGTACAACCGACAT
QCR8	TCACATGGGTGGTCCAAAGC	ACTCGTTACCGTTCTTCCACC
GND2	ACGGATTTACCGTGGTTGCT	TAAAGTGTCGACCGGAGCAC
QCR9	TGCAGGTGCCTTTGTTTTCC	CCTGCAGCTATTCGAGCCTT

## Data Availability

Data are contained within the article.

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
