# Peer review of "Neem Leaf Extract Exhibits Anti-Aging and Antioxidant Effects from Yeast to Human Cells"

_nutrients, 2024, doi:10.3390/nu16101506_

Round 1

Reviewer 1 Report

Comments and Suggestions for Authors

Dear Authors, I have completed my review of your study that focuses on the anti-aging and antioxidant effects of Neem Leaf Extract. Your work contributes valuable knowledge to the field and highlights the potential of natural compounds in pharmacological applications, which is of high relevance and interest. The data presentation is clear and well-structured. Additionally, the use of network pharmacology to elucidate the mechanisms underlying the effects of Neem Leaf represents a substantial strength of this study. Despite these strengths, several areas require revision to meet the journal's standards and to ensure the accuracy and completeness of the manuscript.

Specific comment:

-Line 13: The full name should precede the acronym 'NLE' at its first use in the manuscript. This practice should be consistent across all acronyms used in the document.

- References currently mentioned in the text do not comply with MDPI citation style, which requires them to be enclosed in square brackets (e.g., [3]). Please revise the citations accordingly.

- The current introduction appears too brief. I recommend expanding this section to include more comprehensive information on the state of research in your field, particularly focusing on network pharmacology methods. I would suggest improving the “Discussion” section as well.

- There is a lack of references to studies on human cells in the aims section, yet this is mentioned in the title and body of the work. Please revise it.

- Section 2: The formatting of subheadings within Section 2 should adhere to MDPI guidelines (e.g., 2.1, 2.1.1, etc.).

- Line 69: Please specify what 'HeLa cells' are to enhance the clarity of the experimental model description.

- I strongly suggest adding a subsection on statistical analysis in the "Materials and Methods" section.

- Line 242: The names of polyphenolic compounds should not be capitalized throughout the manuscript, please revise it.

- Line 390: I would suggest adding a supporting reference in this section.

- Consider adding a paragraph on future perspectives in the "Conclusions" section to provide a forward-looking view.

- The supplementary table should be converted from Excel to Word format to maintain consistency with journal requirements.

Reviewer 2 Report

Comments and Suggestions for Authors

The authors have diligently designed the study and performed the different assays laboriously. However, the presentation of the results must be improved.

So, please explain the (interesting and convincing) results more clearly. Please explain the Figure- and Diagram captures more in detail.

E.g., what means IntDen/Cell in Figure 2?

Also the sentence "enhanced oxidative stress response" can be misunderstood as an overshooting stress response.

Comments on the Quality of English Language

Expressions like "NLE enhances the oxidative stress response of human cells" show that the authors have no "semantic feeling" for the English language.

Round 2

Reviewer 1 Report

Comments and Suggestions for Authors

The authors have replied to all the comments and the manuscript now meets the standards for publication.